# Role of Immune Cells and Receptors in Cancer Treatment: An Immunotherapeutic Approach

**DOI:** 10.3390/vaccines10091493

**Published:** 2022-09-07

**Authors:** Anirban Goutam Mukherjee, Uddesh Ramesh Wanjari, Arunraj Namachivayam, Reshma Murali, D. S. Prabakaran, Raja Ganesan, Kaviyarasi Renu, Abhijit Dey, Balachandar Vellingiri, Gnanasambandan Ramanathan, George Priya Doss C., Abilash Valsala Gopalakrishnan

**Affiliations:** 1Department of Biomedical Sciences, School of Biosciences and Technology, Vellore Institute of Technology (VIT), Vellore 632014, Tamil Nadu, India; 2Department of Radiation Oncology, College of Medicine, Chungbuk National University, Chungdae-ro 1, Seowon-gu, Cheongju 28644, Korea; 3Department of Biotechnology, Ayya Nadar Janaki Ammal College (Autonomous), Srivilliputhur Main Road, Sivakasi 626124, Tamil Nadu, India; 4Institute for Liver and Digestive Diseases, Hallym University, Chuncheon 24252, Korea; 5Centre of Molecular Medicine and Diagnostics (COMManD), Department of Biochemistry, Saveetha Dental College & Hospitals, Saveetha Institute of Medical and Technical Sciences, Saveetha University, Chennai 600077, Tamil Nadu, India; 6Department of Life Sciences, Presidency University, Kolkata 700073, West Bengal, India; 7Human Molecular Cytogenetics and Stem Cell Laboratory, Department of Human Genetics and Molecular Biology, Bharathiar University, Coimbatore 641046, Tamil Nadu, India; 8Department of Integrative Biology, School of Bio Sciences and Technology, Vellore Institute of Technology (VIT), Vellore 632014, Tamil Nadu, India

**Keywords:** cancer, immune cells, checkpoints, CART cell, monoclonal Abs, immunotherapy, combination therapy

## Abstract

Cancer immunotherapy moderates the immune system’s ability to fight cancer. Due to its extreme complexity, scientists are working to put together all the puzzle pieces to get a clearer picture of the immune system. Shreds of available evidence show the connection between cancer and the immune system. Immune responses to tumors and lymphoid malignancies are influenced by B cells, γδT cells, NK cells, and dendritic cells (DCs). Cancer immunotherapy, which encompasses adoptive cancer therapy, monoclonal antibodies (mAbs), immune checkpoint therapy, and CART cells, has revolutionized contemporary cancer treatment. This article reviews recent developments in immune cell regulation and cancer immunotherapy. Various options are available to treat many diseases, particularly cancer, due to the progress in various immunotherapies, such as monoclonal antibodies, recombinant proteins, vaccinations (both preventative and curative), cellular immunotherapies, and cytokines.

## 1. Cancer and Immunotherapy

Immunotherapy has transformed cancer treatment by helping the body eliminate malignant cells by boosting the body’s natural defenses. Adoptive cell transfer (ACT), which utilizes autologous immune cells, and immune checkpoint inhibitors (ICIs), which interrupt the coinhibitory signaling pathways, are among the available immunotherapy approaches. The efficiencies of these therapies vary from patient to patient, and only a subset of patients may benefit from them [1]. Cells of the innate immune system, such as eosinophils, neutrophils, basophils, natural killer cells, macrophages, dendritic cells, and monocytes, as well as the innate adaptive system supported by lymphocytes, including T cells and B cells, enter the tumor microenvironment (TME) and effectively control the growth of tumor cells [2,3,4,5,6] (Figure 1).

The field of cancer immunology has seen significant advancements in understanding and research during the past few decades. Our comprehension of the processes and routes that control the immune system’s sensitivity to cancer has been significantly bolstered by discoveries in the past few decades [7]. Immunotherapy is an innovative cancer treatment involving adaptive modification of the immune system to attack cancer cells in multiple targets and directions [8]. In contrast to conventional cancer treatments, such as radiotherapy and chemotherapy, immunotherapy is a recent cancer treatment. The primary goal of immunotherapy is to fortify the patient’s immune system by manipulating the immunological milieu so that immune cells can better attack and eliminate tumor cells at many key nodes [9]. The majority of benefits of immunotherapy are considerably amplified when combined with conventional antitumor therapy or numerous immune checkpoint inhibitors (ICIs), although the optimal circumstances require additional in-depth research [10].

Antitumor immunotherapy has become a pivotal treatment option [11]. Studies involving the treatment of various cancerous growths have produced promising results, and the discovery of novel targets and strategies has increased the effectiveness of immunotherapy while decreasing side effects [12]. Nevertheless, immunotherapy is not without its detractors; for example, therapies can be limited by a lack of empirical validation, some serious side effects and even death, the randomization of therapeutic efficacy, and the considerable expense of medication [10].

“Immunoinflammatory” tumor treatment has a positive effect and considerably boosts patients’ chances of surviving in the later stages of the disease [13]. Immunotherapy is associated with high precision and specificity and maintains its efficacy over an extended period [14]. The medication stimulates the body’s immune system to revive immunological functioning and destroy tumor cells over an extended period [15]. It can restore and enhance the function of the immune system; completely identify, hunt for, and kill tumor cells; and successfully prevent the recurrence and spread of tumors [16]. With careful consideration, medication can completely eradicate any remaining cancer cells and tiny lesions that may be present in the body. However, standard cancer medicine is also associated with unpleasant side effects [10].

The effectiveness of immunotherapy is reduced when the patient has a tumor of the “immune suppression type” or the “immune exclusion type” [17]. Inhibiting immune checkpoints can result in negative regulation, resulting in autoimmune disorders and even death [18]. Some patients may experience a range of nonspecific harmful side effects after using the medication, and they may even develop a hyper-progressive disease with the potential to speed up the dying process [19]. Many factors can influence the effectiveness of immunotherapy. It is impossible to determine how long patients will live or their prognosis, and the price tag for treatment is quite steep [10].

## 2. Dendritic Cells and Cancer

Three features of dendritic cell (DC) biology are related to its anticancer immune responses and how DCs aid in the efficient response of T cells to tumorigenesis. Conventional dendritic cells (cDCs) were previously referred to by various terms, such as CD103^+^ cDCs, CD8α^+^ DCs, or Xcr1^+^ cDCs. However, they are now generalized and called cDC1s, as all the previously used terms indicate the same family of typical DCs [20]. The ‘cross’ term in ‘cross priming’ defined by Bevan refers to the cross-presentation of MHC alleles with endocytic or exogenous antigens for intracellular processing [21]. Under in vitro conditions, cross-presentation can occur between antigens and cell types such as cDC1s, Mo-DCs, and GM-DCs. Cross presentation by Mo-DCs is believed to be related to IRAP, Rab27a, Rab3b/c, Rac2, NOX2, the mannose receptor, Sec22b, Sec61, and TFEB [22].

Some studies show that when ovalbumin (OVA) is soluble or coupled with surface receptors or Fc, specific antibodies are utilized as antigens, and cDC2s were reported to be more efficient than cDC2 in the processing and presentation of MHC-II antigens [23]. cDC2s induced CD4 T-cell proliferation in a tumor model, with antitumor activity produced by the loss of regulatory CD4 T cells. Hence, the primary APCs that stimulate CD4 T-cell responses are cDC2s [24].

## 3. Natural Killer (NK) Cells and Cancer

Cancer cells develop defense mechanisms against the immunity provided by natural killer (NK) cells against cancer. Several immunotherapy approaches based on NK cells are available to overcome NK cell paralysis, such as adoptive cellular immunotherapy, which uses allogeneic NK cells as self-histocompatibility antigens that do not hinder immunity [25]. NK cells are large, granular lymphocytes and exhibit natural cytotoxic effects against cancer cells, even without preimmunization, the most potent of which are CD56dim NK cells, the major circulating cells. The fact that NK cells can eliminate tumor cells was proven by the results obtained in studies conducted in mouse xenograft tumor models [25,26].

NK cells kill the target tumor cells by tumor cell apoptosis through caspase-dependent and caspase-independent pathways caused by the release of perforin and granzymes containing cytoplasmic granules, which are released in a calcium-dependent manner into the intracellular space [27]. The perforin in the cytoplasmic granules induces perforations in the cell membrane, allowing granzymes to enter the tumor cells, thereby resulting in apoptosis mediated by cell death receptors [28]. TNF family members, such as FasL or TNF-related apoptosis-inducing ligands (TRAILs), are expressed by some NK cells, which, upon interaction with their corresponding receptors (Fas and TRAILs), induce apoptosis of tumor cells by limiting tumor angiogenesis and enhancing adaptive immunity through the release of many effector molecules with anticancer properties, such as IFN-γ [25,29]. Nitric oxide is produced as a result of exposure to tumor cells. This signaling is used by NK cells to kill cancer cells by destroying tumor cells via the expression of CD16 [30]. Stimulating cytokines, such as IL-2, IL-12, IL-15, and IL-18, or those that cause the production of IFN, can further increase the antitumor effects of NK cells [31].

### 3.1. Autologous NK Cells

The major strategy to enhance the anticancer activity of NK cells via endogenous NK cell activation and NK cell proliferation is through the timely administration of cytokines, such as IL-2, IL-12, IL-15, IL-18, and IL-21, and type 1 IFNs [31]. After being activated by cytokines, NK cells are transformed into lymphokine-activated killer (LAK) cells. These cells exhibit an enhanced cytotoxic effect against malignant cells by upregulating effector molecules, such as adhesion molecules, NKp44, granzymes, perforin, TRAIL, and FasL, and increasing proliferation and cytokine production [32] (Figure 2).

### 3.2. Allogeneic NK Cells

Enhanced tumor-killing activity and improved control over acute myeloid leukemia (AML) relapse are shown by alloreactive NK cells with KIR mismatch. Based on clinical evidence, specific criteria have been established for selecting mismatched donors [33]. Clinical evidence proves that allogeneic NK cells can be used to control malignancies in humans. Leukemia and solid malignancies can be treated safely and effectively with adoptively transferred human-mismatched allogeneic NK cells [34].

### 3.3. Genetic Modification of NK Cells

Cytokine gene transfer methods increase NK cell proliferation and survival capacity. The modifications of genes such as IL-2, IL-12, and IL-15 have been proven to improve the proliferation rate and enhance the cytotoxic capacity, in vivo anticancer activity, and survival ability [35]. Gene transfer of receptors specific to chimeric tumor antigens is performed in this method to retarget NK cells to tumor cells. Chimeric receptors show enhanced specific cytotoxicity against carcinoembryonic antigen (CEA), HER2/neu, and CD33 in NK cell lines in vivo and in vitro [36]. Furthermore, compared to the control, the chimeric antigen receptor carrying NK-92 cell lines containing fragment CD20-specific scFv Ab showed increased cytotoxic effects against CD20^+^ target cells [37]. Likewise, NK cells transduced with chimeric CD19 receptors show increased cytotoxicity against CD191 malignant B cells [32].

## 4. B-Cell and Cancer

Regulatory B cells (Bregs) play a crucial role in modulating autoimmune, inflammation, and cancer-related immunological responses [35]. These cells secrete anti-inflammatory mediators, such as IL-10, suppress various cell types, and can attenuate antitumor immune responses by promoting the conversion of T cells into regulatory T cells [38]. To date, researchers have mostly focused on T cells in cancer immunology. However, recent studies provide information about B cells’ positive and negative roles in carcinogenesis and tumor growth [39].

## 5. γδT Cell Therapy and Cancer

Cellular stress biomarkers that are upregulated in various cancer types have been confirmed by recent studies, emphasizing the significance of the γδT cell compartment in tumor immunosurveillance, supporting their use in cell therapy [40].

### 5.1. Stimulating γδT Cells In Vivo with Phosphoantigens

The characteristic ability of the V**γ**9Vδ2 T cell subset to respond to prenyl pyrophosphates has been exploited to redirect these cells to tumors by modifying the isoprenoid metabolism in the cancer cells [41]. Amino moiety containing structural analogs of prenyl pyrophosphates called amino pyrophosphates can be used to achieve such modifications [42]. Using these amino moieties results in the accumulation of prenyl pyrophosphate substrates by indirectly inhibiting farnesyl pyrophosphate synthase (FPPS) in the mevalonate pathway [43]. A direct correlation is observed between tumor detection by V**γ**9Vδ2 T cells and zoledronate-induced FPPS inhibition. In the 50 tumor cell lines screened by Idrees et al., the FPPS-inhibiting concentration of zoledronate was lower than that required to stop the proliferation of tumor cells, demonstrating that the activation of T cells was not caused by cell death [44]. The combination of amino bisphosphonates and conventional chemotherapy may be a desirable therapeutic approach for treating cancer-initiating cells, as this combination can make cancerous cells more susceptible to the cytotoxic effects of V**γ**9Vδ2 T cells. In addition to their antineoplastic cytotoxicity, γδT cells can be utilized as a strong anticancer vaccine because they acquire a professional antigen-presenting phenotype upon activation [45].

### 5.2. Redirecting T Cells to Tumors

Before an adoptive transfer of T cells, exogenous TCR of known antitumor specificity is introduced into the peripheral lymphocytes derived from patients to direct T cells to tumors [46]. Transduction of ab TCRs into αβT cells has been used in most TCR gene investigations. However, this method has a risk factor, as ab TCR mispairing may occur between endogenous and exogenous TCR chains, producing receptors with undefined and potentially autoreactive specificities. This problem can be overcome by γδT cells, which prevent mispairing by introducing tumor-reactive ab TCRs [44]. According to a study by Zhao et al., T cells transduced with V**γ**9Vδ2 TCR and modified with ovarian carcinoma or unidentified antigen-specific CDR3d loop exhibited *in vivo* anticancer activity [47]. Therefore, detailed characterization of cancer-specific γδTCRs would likely lead to new studies on their efficiency in TCR gene transduction [44].

## 6. Regulation of Immune Checkpoints

### 6.1. Myeloid and Lymphoid Cells in the Tumor Microenvironment

The extracellular matrix, lymphoid cells, adjacent blood arteries, immune cells, fibroblasts, and the surrounding tumor microenvironment collectively constitute the tumor microenvironment (TME) [48,49]. Within the TME, infiltrating myeloid tissue, MDSC, TAM, TAN, and Tregs play important regulatory roles in immunological checkpoints, promoting tumor growth and altering tumor-infiltrating lymphocyte function. Consequently, these cell types are prospective cancer immunotherapy targets [50].

The TME includes myeloid-derived suppressor cells (MDSCs), which comprise monocytic (M-MDSCs) and polymorphic nuclear MDSCs (PMN-MDSCs). MDSCs primarily decrease the activity of immune cells [51]. To impair immune cell function within the TME, chemokines produced by various cancers aggressively attract MDSCs to primary and metastatic tumor locations [52]. The roles of MDSCs derived from peripheral lymphoid organs are distinct from those of MDSCs developed in the TME. During PD-1 or CTLA-4 blockade therapy, the ongoing destruction of MDSCs by Ab depletion virtually eradicates pre-existing malignancies [53]. PD-1 or CTLA-4 inhibition is linked to a significant increase in the number of MDSCs in circulation. Multiple processes may be at play when MDSCs in the TME regulate the immune system’s production of CTLA-4 and PD-1. First, tumor-associated hypoxia caused by hypoxia-inducible factor 1-α (HIF1-α) promotes PD-L1 expression on the surface of tumor-infiltrating MDSCs, which inhibits T cell function by binding to PD-1 expressed on T cells. [54]. Second, increased expression of arginase-1 (Arg1) and inducible nitric oxide synthase (iNOS) is inversely correlated with coinhibitory checkpoint receptors and ligand expression on MDSCs [55].” MDSC byproducts, such as TGF, IL-10, CCL4, CCL5, and reactive oxygen species (ROS), might stimulate the production of immunological checkpoint molecules on other invading myeloid cells Tregs [56].

The TME also includes important elements such as tumor-associated macrophages and neutrophils (TAMs and TMNs), which play various roles in the formation of tumors. Blood-circulating monocytic precursors are the source of TAMs and TANs [57]. TAMs and TANs have immunosuppressive properties similar to MDSCs and can activate immunological checkpoints and their receptors. Recent research has revealed that PD-1 expression is present in human and mouse TAMs and increases in animal models. TAMs can also effectively stop T-cell activation [58]. Anti-PD-1 antibodies, coupled with PD-1 on T-cell surfaces, can also be quickly captured by PD-1-TAMs. The Fc domain glycan structure and the Fc receptors (FcRs) expressed by TAMs affect the accuracy of anti-PD-1 mAbs. TANs express PD-1 and can increase the expression of cytokines, such as IL-17A, to mediate resistance to PD-1 inhibition [59]. T-regulatory cells (Tregs) play important inhibitory roles in lymphoid cells inside the TME and regulate the production of immunological checkpoint modulators. The key regulators of Treg formation and activity, CD25 and FoxP3, are expressed by CD4^+^ Tregs, a subgroup of CD4^+^ T cells exhibiting a high level of immunological suppression [60]. TGF-β, IL-10, and IL-35 are just a few of the cytokines that can induce the production of Tregs, which can be further broken down into subtypes, such as CD4^+^, CD25^+^, Tr1, and Th3. Tregs can also control checkpoint operations using various mechanisms [61]. Tregs constitutively express the coinhibitory receptor CTLA-4, and B7 is transported to bind to CTLA-4. As a result, the B7/CD28 pathway, which is supposed to promote the maturation of antigen-presenting cells (APC), is severely impeded [62]. IL-2A intake through CD25 can impair Treg-specific CTLA-4 deficiency. By directly downregulating IL-2 gene transcription, FoxP3 can directly upregulate CTLA-4 and IL-2RA gene transcription [63]. Immune inhibitory cytokines released by Tregs, such as IL-10, TGF, and IL-35, may influence how immunological checkpoints indirectly promote tumor growth. Recent research has also shown that Tregs in mice that express PD-1 or TIGIT have more potent immunosuppressive properties than standard Tregs [64,65].

### 6.2. Mediated by Epigenetics

Epigenetic dysregulation, which significantly contributes to cancer growth, influences tumor development. The genetic modification that alters chromatin structure and gene expression without changing the current nucleotide sequence is called epigenetic modulation [66]. DNA methylation and PTMs constitute epigenetic changes (post-translational histone modifications, including acetylation, ubiquitination, and phosphorylation) [67]. According to recent studies on the function of epigenetic alterations in immunological evasion and resistance, epigenetic modulators have been identified as essential mechanisms to enhance immune responses in the TME and restore immune surveillance and homeostasis. These insights offer a solid foundation for studies utilizing immune checkpoint blockage and epigenetic medication combinations for cancer treatment [68].

### 6.3. Mediated by Gut Microbiota

The human gut is home to more than 100 trillion bacteria, and checkpoint blockade therapy has recently been linked to some microbes. There is a wealth of evidence linking changes in gut microbiota composition to various complex disorders, emphasizing cancer [69,70]. Different microbiota or microbiota-derived substances have been shown to influence immunological responses, including Treg and T-helper17 cells [71]. 16S RNA sequencing in JAX mice showed the synergistic effects of *Bifidobacterium*, with improved tumor control resulting from anti-PD-1 therapy [72].

## 7. Activatable Cancer Immunotherapy in Response to Internal Stimulus

Due to their rapid growth, metabolism, maturation, migration, and metastasis, tumor tissues typically constitute a more acidic microenvironment than normal tissues, as well as somewhat increased levels of oxidation and reduction, increased hypoxic state, and overexpressed enzymes [73,74,75,76]. Additionally, tumor cells have been characterized by six major properties that set them apart from normal cells. These characteristics include maintaining signaling; blocking inhibitors; and triggering angiogenesis, metastasis, replicative immortality, and cell death [77]. The internal stimuli associated with some tumor microenvironments characteristics, such as redox potential, acidic pH, overexpressed enzymes, and hypoxia, have been exploited for activatable cancer immunotherapy [78].

### 7.1. Redox-Activated Immunotherapy

Generally, tumor tissue cells have a higher redox state than normal tissue cells. These factors upregulate SOD, ROS, glutathione disulfide (GSSH), GSH, and thioredoxin. These factors help to activate drug biomaterials used in cancer treatment [78]. GSH has an increased reductive capability in cellular metabolites and plays an important role in redox homeostasis. On the other hand, GSH is involved in maintaining the protein fold by mediating and cleaving disulfide bonds. GSH concentration in cancer cells is two-fold higher than in normal cells. Therefore, GSH acts as a trigger of cancer immunotherapy [79].

#### 7.1.1. GSH-Mediated Activation of Immunological Adjuvants

Other immune adjuvants besides Indoleamine 2,3-dioxygenase (IDO) inhibitors may be used in activatable cancer immunotherapy. A small-particle TLR7/8 agonist of GSH for increased activation of immune cells was developed as a CPG nanotherapeutic by incorporating the gold nanorods in CPG with GSH, forming gold thiol bonds and leading to the activation of cancer immunotherapy [80]. In another recent study, oxaliplatin, a prodrug activated by light with photosensitizer pheophorbide A, was integrated with GSH as a heterodimer molecule, improving cancer immunotherapy [81]. Light from the external source stimulates the GSH internal self-assembled nanoparticles to -inactivate cancer cells. TEM and DLS showed how nanoparticles changed morphologically in response to light and GSH activation. This approach can also revert the immunosuppressive tumor IDO-1 and CTL responses [82].

#### 7.1.2. GSH-Mediated Activation of Antigens

Cancer-associated antigens can be used to create immunoactivities vaccines. Owing to the unique tumor microenvironment, GSH-responsive biomaterials can also be used for antigen delivery and activation of tumor locations [78]. Moon and colleagues developed homogeneous, biodegradable, mesoporous silica nanoparticles for neoantigen-based cancer vaccination (bMSN) [83]. These bMSN nanoplatforms can be loaded with a variety of neoantigen peptides, CpG oligodeoxynucleotide adjuvants, and the photosensitizer chlorin e6 for combined cancer treatment (Ce6) [78]. To specifically target tumors, these neoantigen peptides were disulfide-bonded to the surface of bMSN, and the highly abundant GSH present in thetumor’s intracellular milieu was able to cleave them further [84]. Strong neoantigen-specific CD8^+^ CTL responses could be induced for individualized cancer immunotherapy via neoantigen release and PDT-mediated DC recruitment [85].

#### 7.1.3. GSH-Mediated Activation of Immunotherapeutic Antibodies

Immunotherapeutic antibodies, in addition to immunological adjuvants and tumor-associated antigens, can generate potent antitumor immunity via GSH-mediated release [86]. Researchers developed cell-surface-conjugated protein nanogels (NGs) that respond to an increase in T-cell surface reduction potential after antigen recognition to limit medication release to areas of antigen contact [87]. NGs were then conjugated on the surface of T cells with an interleukin-15 super-agonist (IL-15Sa) combination. Once T cells were activated, the T-cell surface reduction potential increased, resulting in drug release and increased stimulation of these T cells [88].

Tumor cells typically have a substantial GSH concentration due to their aberrant cellular metabolism and oxidative stress [89]. GSH-activatable characteristics are important for the prevention of tumorigenesis. A GSH-activatable nanosystem with tumor invasion ability was developed to achieve extremely effective immunotherapy. GSH-activatable medication delivery is the most effective approach to improve ICD and reverse the immunosuppressive tumor microenvironment (ITM) [90]. GSH is essential for cellular development and division, drug metabolism, and free radical removal. The GSH concentration in cancer cells is significantly greater than that of healthy cells. The diminution of glutathione (GSH) is an effective way to boost the effectiveness of other cancer treatments [79].

### 7.2. Enzyme-Activated Immunotherapy

Numerous malignancies have been found to improperly express different enzymes that are essential for the development of tumors. Enzymes may catalyze the breakdown of related substrates with considerable selectivity and efficiency and typically exhibit unique activities and functions [91]. As a result, numerous enzyme-specific biomaterials have been created for specialized cancer cell diagnosis. HAase, caspase, and metalloprotease (MMP), were used to improve anticancer immunity [92].

#### 7.2.1. MMP-Activated Immunotherapy

A combinational nanosystem for MMP-responsive drug boost and anticancer immunotherapy combined with the photosensitizer indocyanine green (ICG) with an inhibitor PD-L1 [93,94] involving self-assembly of a PD-L1, ICG, (d)-epigallocatechin-3-O-gallate dimer (dEGCG), and an MMP-2-liable PEGylated dEGCG resulted in the formation of a PEG-PLGLAG-dEGCG [78]. The activity of PD-L1 was efficiently shielded in the nanoparticles, and PD-L1 was reactivated after MMP-2 or Triton X-100 disassociated the nanoparticles [95]. Combining MMP-2-responsive prodrug molecules with anticancer agents such as adjudin and cisplatin with peptide FPR-1 can inhibit cancer progression [96]. MMP-2 overexpression in tumors can cause localized activation of immunological and chemotherapeutic medications to kill tumor cells and evoke potent anticancer immunity for improved immunotherapy and chemotherapy [97].

Due to their involvement and upregulation in several malignancies, matrix metalloproteinases (MMPs) are a well-studied class of secreted and membrane-bound proteases. A total of 71 tumors with elevated MMP-2/-9 capabilities were targeted as stimuli owing to their ability to catabolize signals and spread them throughout the body [98]. Multiple MMPs (matrix metalloproteinases) play roles in the metastatic cascade. By releasing MMP9, MMP10, and MMP15, tumor cells can invade neighboring tissues, break through the basement membrane, and move throughout the extracellular matrix [99]. Disruption of the arterial basal lamina by matrix metalloproteinases 2, 9, and 14 (MMP 2, 9, and 14, respectively) facilitate tumor cell intravasation and extravasation [99,100]. ECM degradation and the release of proangiogenic factors, such as VEGF, FGF-2, and TGF-, MMP1, 2, 7, 9, and 14 control angiogenesis [101]. Furthermore, the efficiency of MMPIs can be improved through targeted distribution using MMP-activated prodrugs or MMP-degradable drug carriers, such as nanoparticles or hydrogels [99,102,103]. MMPs are not overexpressed in abnormal cells and exhibit less immunogenicity than wild-type toxins [104].

#### 7.2.2. Caspase-Activated Immunotherapy

Protoporphyrin IX (PpIX), a photosensitizer, and 1-methyltryptophan (1MT), an IDO inhibitor, were combined to create a chimeric peptide using the peptide linker Val-Asp Asp-Glu-(DEVD) [105]. These peptides may passively collect at tumor locations and self-assemble into PpIX-1MT nanoparticles. Nanoparticles produce ROS upon 630 nm light irradiation, causing tumor cells to undergo apoptosis [106,107], aiding in the production of caspase-3 and prompting the release of 1MT to counter immunosuppressive tumor microenvironments and activation of anticancer immune responses [108]. This cascade synergistic anticancer approach, which combines PDT with activation of the immunological drug 1MT in response to caspase-3, offers a workable method for activatable cancer immunotherapy. Chemotherapy was combined with this caspase-3-activated cancer immunotherapy approach for improved antitumor treatment [109]. Doxorubicin (DOX)-encapsulated mesoporous silica nanoparticles that had undergone iRGD modification were combined with 1MT by peptide linker DEVD [110]. Once absorbed by tumor cells, DOX may be released to cause cell death and the production of caspase-3. After that, the DEVD peptide sequence’s cleavage might release 1MT in a cascade that would activate the immunesystem’s ability to fight off tumors and have a greater therapeutic impact [105].

DNA degradation and eventual cell death are brought about by the recruitment of FADDs and activation of caspase-8 via homotypic interaction of the death effector domain (DED) [111]. Apoptosis-inducing factor (AIF) and endonuclease G (EndoG) are both found in mitochondria and mediate cell death in a caspase-independent manner [112,113]. It has been demonstrated that granzyme B can trigger apoptosis by setting off a caspase-3 amplification cascade in the mitochondria [114]. In a similar spirit, researchers found that releasing proapoptotic molecules from the mitochondria is necessary for granzyme B to activate caspase-3 [115]. DNA fragmentation can be induced by granzyme B, even in the absence of active caspases, by cleaving the nuclear caspase substrate inhibitor of caspase-activated DNase (ICAD), generating CAD and thereby bypassing the necessity for caspases [116,117,118,119]. Granzymes, a family of serine proteases, represents a key element of these cytotoxic granules. Granzymes stimulate apoptosis by activating caspases and directly proteolyzing intracellular substrates, such as lamin B; α-tubulin, an inhibitor of caspase-activated DNase (ICAD); BH3-only protein (BID); and DNA-dependent protein kinase (DNA-PK) [120,121].

#### 7.2.3. Hyaluronidase (HAase)-Activated Immunotherapy

The combination of a PD-L1, polylysine, and 1M-conjugated and Ce6-conjugated hyaluronic acid (HA) resulted in the creation of a nanoplatform. [122]. Due to HAase’s breakdown of heavy chain (HC) and subsequent release of PD-L1 after 4 h of incubation, the particle size was reduced [123]. Following systemic injection, tumor extracellular matrix (ECM)-overexpressed HAase broke down the nanoparticles, releasing a PD-L1, followed by inhibition of IDO by 1MT for improved immunoactivities [124].

Hydrolysis of hyaluronic acid is catalyzed by the HAase family of endoglycosidases (HA). Malignant melanoma, bladder cancer, and prostate cancer all exhibit HAase overexpression [125,126]. Thanks to the enzymatic breakdown of HA, which is overexpressed in the tumor microenvironment, a microneedle-based platform could improve local retention and promote the release and activation of immunotherapeutic drugs [122]. The potential of these nanoprobes to inflict harm on tumor cells in real-time is encouraging because HAase is overexpressed in these cells [127].

## 8. Immunotherapy for Cancer

The human immune system protects against infections and foreign pathogens. The link between cancer and the human immune system has been the subject of numerous preclinical and clinical studies [128]. Innate and adaptive immune systems have been found to work together to overcome cancer. Cancer immunity mainly relies on CD8^+^ cytotoxic T lymphocytes (CTLs); tumor-specific CTLs proliferate and are directed to tumor sites, where they attack cancer cells due to professional APC (pAPC) cross-priming of naive CD8^+^ T cells [129]. Recent developments in cancer immunology seek to block the regulators of immune checkpoints, overcome immune tolerance, or identify novel tumor antigens through next-generation sequencing [128] (Table 1).

### 8.1. Monoclonal Antibodies

The use of monoclonal antibodies in immunotherapy has become an important component in cancer therapy, in addition to radiation [139], chemotherapy, and surgery, as they possess numerous mechanisms of action that are clinically relevant [140]. Antibody-modified proteins directly target tumor cells, inducing long-lasting antitumor immune responses and interfering with cancer’s immunological transduction pathways. Nowadays, naked, conjugated, and bispecific mAbs are used in cancer treatments [141]. The first approved mAb is a naked, non-conjugated, chimeric mAb, rituximab, which targets the CD20 antigen in treating B-cell non-Hodgkin’s lymphomas [142]. An example of bispecific mAbs is blinatumomab, which is used to treat acute lymphocytic leukemia and binds to CD3 and CD19 [143]. Elotuzumab is the first humanized and signaling lymphocytic activation molecule family 7 (SLAMF7) member targeting mAb. These mAbs work by attaching and blocking the tumor cells’ antigens, increasing the immune response against cancer cells and destroying them [144].

### 8.2. Bispecific Antibodies

Bispecific antibodies (BsAbs) were developed because diseases have multiple causes. Four areas were targeted in developing BsAbs: suppression of two cell surface receptors; blockage of two ligands; cross-linking of two receptors; and recruitment of T cells, which typically lack an Fc receptor and are not activated by antibodies [145]. BsAbs were first created by the reduction and reoxidation of hinge cysteines in monoclonal antibodies or by the union of two hybridoma cells to create hybridomas or quadroma cells. BsAbs can be categorized based on their function, either utilizing their specificity for targeted delivery of a toxin or any other therapeutically active compound or directly activating and neutralizing their targets [146,147].

### 8.3. Cancer Vaccine

Therapeutic and preventive vaccinations that makeup cancer vaccines operate as response modifiers to boost or re-establish the immune system’s capacity to combat cancer. Therapeutics aims to immunize patients against tumor-specific or tumor-associated antigens to stimulate antitumor T cells [148]. In contrast, cancer vaccines act by initiating an attack against cancer cells. To increase the effectiveness of vaccines, they are mostly administered with adjuncts called adjuvants [149]. One such natural adjuvant is dendritic cells. DC vaccination is performed by directly targeting antigens to the DC receptors or by producing antigen-loaded DCs ex vivo [150]. Sipuleucel-T is used for the treatment of metastatic castrate-resistant prostate cancer. Therapeutic cancer vaccines have emerged as an attractive approach to inducing long-term antitumor immunity [151]. Human papillomavirus (HPV) vaccines and hepatitis B virus (BBV) vaccines have been approved by the FDA; these vaccines use tumor antigens, peptides, or entire cancer cells to trigger the immune system [152].

### 8.4. Adoptive Cell Therapy

As an attractive form of immunotherapy against solid cancers and hematologic malignancies, adoptive cell therapy (ACT) involves preventing or treating a disease by administering immunologically active cells to patients [153]. An alternative approach to this method is the transfer of antigen-specific TCR genes through T-cell transduction with either lentiviruses or retroviruses into the lymphocytes isolated from the patient’s peripheral blood [154]. The second type of modified T cells is CAR-modified T cells. CD19 on B-cell malignancies can be targeted with CAR-expressing T (CAR-T) cells and has proven incredibly effective [155].

### 8.5. Molecular Chaperones and Cancer Immunotherapy

The molecular identification of a large number of antigens that are linked with tumors has offered targets for the creation of new immunotherapies for the treatment of cancer [156]. The role of molecular chaperones in tumor immunity and the useful features of molecular chaperones in cancer therapy has garnered increasing interest in recent years [157,158]. Chaperones can be classified according to to the following three tenets: The first advantage is that chaperones can bind to a wide variety of peptides and proteins connected with the tumors [159]. The second factor is the presence of particular receptors on the surfaces of antigen-presenting cells (APCs), which enables the effective uptake of chaperones combined with peptides and proteins [160,161]. Thirdly, chaperones interact with and activate innate immune components (such as APCs or NK cells), which assists in the initiation of adaptive immunological responses (such as the activation of CD8^+^ CTL and CD4^+^ T helper cells) [162,163]. Heat-shock proteins, often known as HSPs, are a type of molecular chaperone that is evolutionarily conserved but plays various roles in various physiological processes [164]. HSPs are categorized according to molecular size into the following groups: HSP27, HSP40, HSP60, HSP70, and HSP90 [165]. By preserving the native folding energetics of the proteins, HSPs inhibit the nonspecific aggregation of proteins within the cell. HSPs are effective biomarkers for the stage of some types of cancer and the severity of the disease [166]. HSPs have been shown to play a role in the proliferation of tumor cells and their differentiation, invasion, and metastasis [167]. The expression levels of HSP27 and HSP70 in tumor cells were found to affect how well the cells responded to traditional cancer treatment [168]. Another important heat shock chaperonin protein, HSP60, is mostly found in mitochondria, where it contributes to the folding and transport of mitochondrial proteins [169]. HSP60 is a heat-shock chaperonin protein. The high-molecular weight HSP90 chaperone is a crucial regulator of the process of tumor growth, similar to the other HSPs [170]. Not only are HSPs involved in the growth of tumors, but they also play a role in determining how they react to treatment. An increasing body of evidence demonstrates that HSPs can be effective targets for cancer immunotherapy [167,171]. SRECI binds to a much wider variety of common heat-shock proteins than LOX-1, including HSP60, HSP70, HSP90, HSP110, gp96, and GRP170 [172]. The majority of LOX-1 binding occurs with HSP60 and HSP70. It is essential for immunosurveillance for peptides to be cross-presented in this manner because not only is the attached peptide protected from degradation, but the efficiency of cross-presentation is also increased in dendritic cells [173]. Some heat-shock proteins, such as HSP70 and HSP90, are also implicated in the intracellular cytosolic pathway of cross-presentation and the transportation of antigens from the endosome into the cytosol [163]. This process allows antigens to be transported from the endosome into the cytosol. Glucose-regulated protein 94 (GRP94), also known as GP96, is a heat-shock protein (HSP) family member that acts as a stress-inducible molecular chaperone [174]. GRP94 plays an essential role in the regulation of the delicate balance that between the survival and death of cancer cells [175]. Additionally, GRP94 is important for the chaperoning of many proteins, some of which have been shown to play essential roles in immunological response and in the genesis of cancer [158]. GRP94 protein has significant potential as both a biomarker and a therapeutic target. It has come to light that GRP94 is involved in the process of survival signaling by way of its client protein network, as well as the induction of UPR and the modulation of the immune response. There is still potential for a targeted treatment that involves the selective suppression of GRP94. To date, the GRP94 client network has not been elucidated in its entirety [176].

## 9. Combination Therapy for Cancer

Preclinical and clinical trials of combination therapies of mAB with radiation therapy; chemotherapy; molecular target drugs, like tyrosine kinase inhibitors or vaccines; and other antibodies against the same target or cellular therapies are now being conducted [177]. Tumor cells are destroyed by tumor-cell-specific proteins released by the NK-cell-mediated ADCC. Presentation to cytotoxic cells by antigen-presenting cells results in an effective antitumor response [178]. Evidence suggests that cetuximab can be combined with anti-PD-1/PD-L-1 mAbs. Immune checkpoint blockers (ICB) are hypothesized to act synergistically with cetuximab, and multiple combinations of ICB synergistically enhance T-cell responses [179].

### CART Cell Therapy for Cancer

Chimeric antigen receptor T (CAR-T)-cell therapy is an innovative treatment method for hematological malignancies. The CAR-T cells are derived from peripheral blood cells that have been genetically engineered [180]. T cells express CAR after the cDNA is integrated into the target cell genome. CAR genes are commonly transduced in T cells using lentiviruses, retroviruses, or other means [154]. Tumor antigen receptors recognize proteins, glycoproteins, and other components, but signaling domains primarily promote T-cell differentiation and proliferation [180]. There is a lack of tumor-specific antigens in solid tumors, and the immunosuppressive nature of the tumor microenvironment makes targeting them with CAR-T cells difficult. This challenge can be overcome by programming T cells with gene modules, improving their therapeutic efficiency and specificity [181]. Signal transmission is overly simplistic and ineffective in the intracellular CD3ζ signaling module originally present in CAR. CARs of the first generation are capable of recognizing tumor antigens and elevating the antitumor activity in T cells, but their low proliferation ability makes them unsuitable for *in vivo* use [182,183]. Second-generation CARs exhibit remarkable cell multiplication and senescence improvements via integrating of CD28 or 4-1BB domains with CD3ζ molecules [184]. CD28 and CD137 costimulatory signals are present in the third generation of CARs [185].

## 10. Conclusions

In recent years, cancer immunotherapy has blossomed into reality thanks to advances in multiple forms of treatment, including cancer vaccines. CAR-T cell therapy can be an effective therapy to eliminate hematologic tumors. Adoptive cell therapy has shown promising results in chronic lymphocytic leukemia (CLL) and some refractory diffuse large B cell lymphoma (DLBCL). As the cellular basis for cancer immunotherapies, tumor-infiltrating immune cells, particularly T cells, must be better understood so that mechanisms of immunotherapies can be deciphered, predictive biomarkers can be identified, and therapies can be more effectively delivered [1].

Immunotherapy is an exciting new type of advanced cancer treatment that has the potential to represent a major improvement in the management of tumors. Future research should focus on restoring specific immunosuppressive pathways in the antitumor process rather than simply enhancing the broad and untargeted systemic immune response. Such research should consider the following three principles: deciding that the tumor causes the immunosuppressive microenvironment, focusing immunosuppression on the tumor microenvironment, and finding new targets acting on the main functional pathways [10,186]. Therefore, cancer immunotherapies aim to clarify the molecular and cellular mechanism by which cancer cells can dodge the immune system, resulting in therapeutic interventions that increase antitumor immunity [187]. Cancer immunotherapies are effective in treating patients with advanced stages of the disease. We anticipate that upcoming development in cancer immunotherapy will overcome and address a considerable number of such problems. The development of more therapeutic targets, personalized biomarker profiles, drug combination therapies that strengthen efficacy and reduce toxicity, and immunopreventive techniques that will reduce cancer rates, relapse, and related treatment costs, are expected to be among the anticipated innovations [7].

## Figures and Tables

**Figure 1 vaccines-10-01493-f001:**
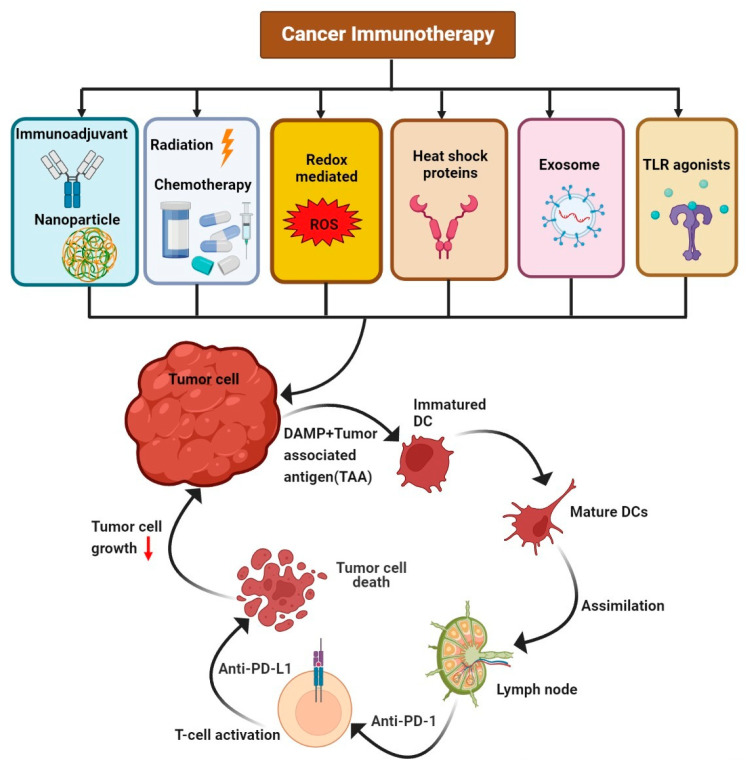
The various available immunotherapy treatments, such as immunoadjuvants with nanoparticles, radiation, chemotherapy, redox-mediated therapy, heat-shock proteins, exosomes, and TLR agonists, that act upon tumor cells, leading to the activation of T cells and suppressing tumor cell growth.

**Figure 2 vaccines-10-01493-f002:**
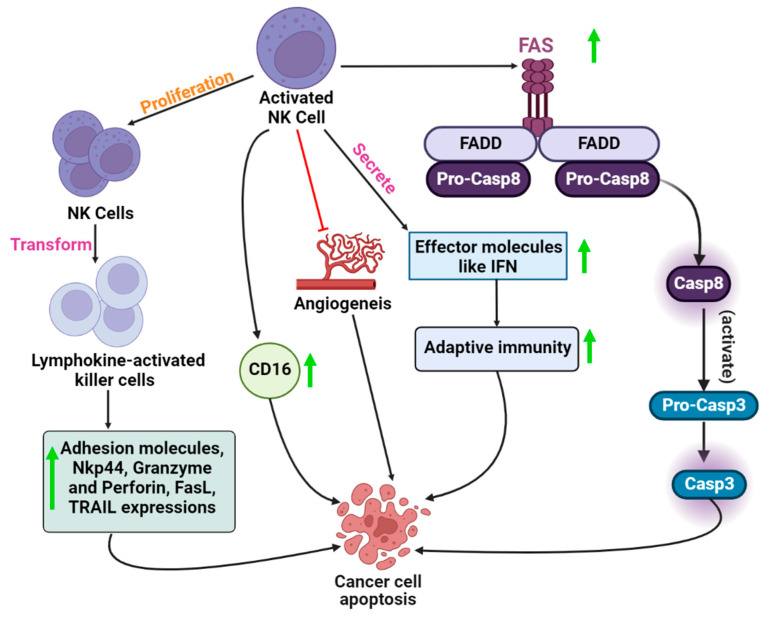
The mechanistic role of activated NK cells in cancer cell apoptosisincludes the activation of the Fas-mediated signaling pathway, inhibition of angiogenesis, high expression of CD16 cytokines, and transformation of NK cells to lymphokine-activated cells.

**Table 1 vaccines-10-01493-t001:** Immunotherapies for cancer treatment.

Cancer Type	Immunotherapy Approach	Targets/Mechanism	Study Type	Cell Lines Used	Animals Used	Technologies Used/Assays Performed	Results	Reference
**Bladder cancer**	**Monoclonal antibodies**							
	(i) KMP1 mAb	KMP1 binds to CD44 and blocks its function	In vivo and in vitro	Human bladder cancer cell lines EJ, BIU-87, and T24; normal human bladder cell line HCV29; human liver cancer cell line HepG; human cervical cancer cell lines HeLa	BALB/c normal mice and nude mice	Mass spectrometry and antigen affinity to determine KMP1 mechanism; RNA interference technology to knockdown CD44 expression	Bladder cancer clinical severity and prognosis were consistent with the expression of KMP1 epitope	[130]
	(ii) R3 mAb/ vofatamab	Inhibits proliferation and FGFR3 signaling by binding to wild-type FGFR3 and FGFR3 mutants	In vivo and in vitro	RT112, RT4, OPM2, Ba/F3, and UMUC-14	Female *nu/nu* mice or CB17 SCID mice	Cell proliferation assay, FACS assay, clonal growth assay, and FACS assay	Bladder cancer development was reduced in vivo by induced shRNA knockdown of FGFR3.	[131]
	**Immune checkpoint inhibitors**							
	(i) Atezolizumab	Blocks immune checkpoint PD-L1/PD-1; reduces immunosuppressive signals; increases T-cell-mediated immunity against tumors	In vivo In vivo and in vitro	Human bladder cancer cell line pumc-91			In comparison to historical controls treated with conventional second-line regimens, individuals with advanced bladder cancer treated with atezolizumab exhibited a significantly improved response rate and survival	[132]
	**Bispecific antibody anti-CD3 x anti-CD155**	CD155Bi-Ab-armed ATCs secrete more IFN-γ and TNF-α, which increases cytokines and activates endogenous immune cells in vivo, inducing an immune response against tumor cells		MBT-2 cell line		Flow cytometry and ELISA	For CD155-positive bladder cancer, CD155 is a useful target. Additionally, CD155Bi-Ab-armed ATCs show promise concerning developing a novel approach to the present treatment of CD155-positive bladder cancer.	[133]
	**BCG vaccines**	Prominent infiltration of the bladder wall by immunocompetent cells and the release of cytokines into the urine	In vivo		Female C3H/HeN mice		Vaccine increased NK cell activity	[134,135]
**Breast cancer**	**Monoclonal antibodies**							
	Trastuzumab	Inhibits intracellular signaling by binding to the extracellular domain of the receptor	In vitro	SK-BR-3 cell line			NK cells killed trastuzumab-coated erbB2-overexpressing cells through an ADCC mechanism mediated by the FcRIII receptor (CD16)	[136]
**Lung cancer**	**Monoclonal antibodies**							
	Cetuximab	Binds to the extracellular domain of EGFR and blocks EGFR-mediated signal transduction	In vitro	LK-1, EBC-1, A549, LK87, Lu99, N417, Ms1, and LU65; epidermoid carcinoma cell line (A431)		Flow cytometry and immunohistochemistry	A correlation was observed between EGFR molecules on the cell, exerting cytotoxicity against lung cancer cell lines.	[137]
	**Monoclonal antibodies**							
**Prostate cancer**	BLCA-38	In patients with prostate cancer, the BLCA-38 antibody binds primarily to prostate cancer cells but not to normal cells and may be useful in targeting novel therapies	In vivo and In vitro	PZ-HPV-7 prostate cells; LNCaP, DU145, and PC-3; LNCaP-C4 and LNCaP-C4–2; LNCaP-LN3, PC3-M, and PC3-M-MM2; MDA PCa 2a and MDA PCa 2b; LAPC4 cells.	Male 6–8-week-old athymic nude mice, BALB/c (nu/ nu)	Flow cytometry	Prostate cancer lines PC-3, PC-3 M, PC-3 M-MM2, and DU-145 all expressed cell surface BLCA-38 antigen, whereas LNCaP, MDA PCa 2a, and MDA PCa 2b did not.	[138]

## Data Availability

Data are available from the authors upon request (A.V.G.).

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
