# Peer review of "Role of Immune Cells and Receptors in Cancer Treatment: An Immunotherapeutic Approach"

_vaccines, 2022, doi:10.3390/vaccines10091493_

Round 1
Reviewer 1 Report
The authors wrote a comprehensive review of immunotherapy and its background in cancer. The review sheds light on all aspects of this topic. There is nothing to add.
I would like to propose two changes that would improve readability and comprehensibility:
(1) The structure with many sub-points makes reading confusing. A table of contents on the first page would be helpful (decision of the journal)
(2) A figure "immunotherapy" showing the influence of the various immune cells and receptors on cancer treatment would display the topic to the reader in a clearer and thus more understandable way.
Author Response
Reviewer 1
The authors wrote a comprehensive review of immunotherapy and its background in cancer. The review sheds light on all aspects of this topic. There is nothing to add.
I would like to propose two changes that would improve readability and comprehensibility:
(1) The structure with many sub-points makes reading confusing. A table of contents on the first page would be helpful (decision of the journal)
Response 1: Authors are thankful to the reviewer for the advise. We have satisfied the comments by adding the table of contents (TOCs) in the main manuscript, highlighted in Yellow (Line No. 43-75).
Table of contents:
- Cancer and immunotherapy
- Dendritic cells and cancer
- Natural Killer (NK) cells and cancer
3.1 Autologous NK Cells
3.2 Allogeneic NK Cells
3.3 Genetic modification of NK Cells
- B-cell and cancer
- γδT-cell therapy and cancer
5.1 Stimulating γδT cells in vivo with phosphoantigens
5.2 Redirecting T cells to tumors
- Regulation of Immune Checkpoints
6.1 Myeloid and Lymphoid Cells in the Tumor Microenvironment
6.2 Mediated by Epigenetics
6.3 Mediated by Gut Microbiota
- Activatable cancer immunotherapy in response to internal stimulus
7.1 Redox-activated immunotherapy
7.1.1 GSH-mediated activation of immunological adjuvants
7.1.2 GSH-mediated activation of antigens
7.1.3 GSH-mediated activation of immunotherapeutic antibodies
7.2 Enzyme-activated immunotherapy
7.2.1 MMP-activated immunotherapy
7.2.2 Caspase-activated immunotherapy
7.2.3 HAase-activated immunotherapy
- Immunotherapy for cancer
8.1 Monoclonal antibodies
8.2 Bispecific antibodies
8.3 Cancer vaccine
8.4 Adoptive cell therapy
8.5 Molecular chaperones and cancer immunotherapy
- Combination therapy for cancer
9.1 CART cell therapy for cancer
- Conclusion
(2) A figure "immunotherapy" showing the influence of the various immune cells and receptors on cancer treatment would display the topic to the reader in a clearer and thus more understandable way.
Response 2: We have satisfied the comments by adding the figure, showing basics of immunotherapy for cancer treatment in the main manuscript with track changes. (Line No. 89-94 ; 186-191)
Figure 1: The figure shows the various Immunotherapy Treatments like immunoadjuvant with Nanoparticle, Radiation, chemotherapy, Redox Mediated, Heat-shock proteins, Exosome, TLR Agonists those acts upon the Tumor cell then activation of T cells takes place with the Anti-PD-1 leads to tumor cells death with AntiPD-L1and suppresses tumor cell growth.
Figure 2: This figure illustrates the mechanistic role of the activated NK cell in cancer cell apoptosis includes the activation of the Fas mediated signaling pathway, inhibiting angiogenesis, high expression of CD16 cytokines and transformation of NK cells to lymphokine activated cells.

Reviewer 2 Report
This research article entitled “ Role of immune cells and receptors in cancer treatment: An immunotherapy” provides information on immunotherapy for cancer treatment. The article also highlights immune cells and their role in cancer control. The article has many grammatical and sentence errors, and the language organization needs to be improved. For these reasons, I conclude that the paper should undergo minor revision
1. Please make sure that all keywords have been used in the abstract and the title.
2. Authors may provide greater details of other cancer therpy with their merits and demerits. How immunotherapy is better than others
3. Good introduction, Still, it should be improved. To make the introduction more substantial, the author should provide several updated recent references to substantiate the claim made.
4. Authors can add a section on Molecular chaperones and cancer immunotherapy
https://doi.org/10.1007/3-540-29717-0_13
https://doi.org/10.1155/2019/3267207
https://doi.org/10.3389/fonc.2021.629846
5. Typographical errors can be avoided. The language and grammar used throughout the manuscript need to be improved. Specific attention needs to be given to this which will improve the standard of the manuscript.
6. Please improve the conclusion with clear Future perspectives and strategies with more emphasis on immunotherapy and other upcoming cancer therapy.
Author Response
REVIEWER 2
This research article entitled “ Role of immune cells and receptors in cancer treatment: An immunotherapy” provides information on immunotherapy for cancer treatment. The article also highlights immune cells and their role in cancer control. The article has many grammatical and sentence errors, and the language organization needs to be improved. For these reasons, I conclude that the paper should undergo minor revision
- Please make sure that all keywords have been used in the abstract and the title.
Response 1: All the keywords are used in the abstract, and title content a broad word which summerises all the kywords.
- Authors may provide greater details of other cancer therpy with their merits and demerits. How immunotherapy is better than others
Response 2: We have satisfied the given comment by adding the efficacy of immunotherapy campare to other therapy, and merits and demerits for cancer therapy in the introduction with track changes (Line No. 109-135).
Anti-tumor immunotherapy has emerged in recent years as a pivotal treatment op-tion [11]. Studies of treating various cancerous growths have produced promising re-sults, and the discovery of novel targets and strategies, such as standard treatment, has increased the effectiveness of immunotherapy while decreasing side effects [12]. Never-theless, immunotherapy is not without its detractors; for example, there are therapies with blindness, empirical and limiting; some instances of serious side effects and even death; the randomization of therapeutic efficacy; and the high expense of medication [10].
The “immunoinflammatory” tumor treatment has a positive effect and considerably boosts patients’ chances of surviving to a later stage of the disease [13]. Immunotherapy with a high degree of precision, specificity, and targeting maintains its efficacy over an extended period of time [14]. The medication stimulates the immune system of the body in order to revive immunological functioning and destroy tumour cells for an extended period of time [15]. It has the ability to restore and enhance the function of the immune system, completely identify, hunt for, and kill tumour cells, and it has the ability to suc-cessfully prevent the recurrence and spread of tumours [16]. With careful consideration, It is able to completely eradicate any remaining cancer cells and tiny lesions that may be present in the body. The standard medicine has more unpleasant side effects than this one [10].
There are restrictions placed on the therapy objects, and there is a rigorous selection process for patients. The effectiveness of immunotherapy is reduced when the patient has a tumour of the “immune suppression type” or the “immune exclusion type” [17]. Inhib-iting immunocheckpoints can result in negative regulation, which can result to autoim-mune disorders and even death [18]. Some patients may experience a range of non-specific harmful and side effects after using the medication, and they may even de-velop a hyperprogressive disease, which has the potential to speed up the dying process [19]. There are a lot of different things that can influence how immunotherapy works. There is no way to know how long patients will live or what their prognosis will be. The price tag for treatment is quite steep [10].
- Good introduction, Still, it should be improved. To make the introduction more substantial, the author should provide several updated recent references to substantiate the claim made.
Response 3: We have improved the introduction (Cancer and Immunotherapy) part as per given comments with latest references, track changed in manuscript (Line No. 96-108).
The field of cancer immunology has seen significant advancements in understanding and research during the past few decades. Our comprehension of the processes and routes that control the immune system’s sensitivity to cancer has been significantly bolstered by the discoveries that have been made in the course of study over the course of the past few decades [7]. Immunotherapy is a creative treatment for cancer that adaptively modifies the immune system to attack cancer cells in multiple targets and directions [8]. In contrast to conventional cancer treatments such as radiotherapy and chemotherapy, immunotherapy is a more recent form of cancer treatment. Immunotherapy’s primary goal is to fortify the patient’s immune system by manipulating the immunological milieu in such a way that the patient’s immune cells are better able to attack and eliminate tumour cells at many key nodes [9]. The majority of the benefits will be greatly amplified when used in conjunction with conventional anti-tumor therapy or numerous immunocheckpoint inhibitors (ICIs), although the specific circumstance still has to be researched in greater depth [10].
- Authors can add a section on Molecular chaperones and cancer immunotherapy
https://doi.org/10.1007/3-540-29717-0_13
https://doi.org/10.1155/2019/3267207
https://doi.org/10.3389/fonc.2021.629846
Response 4: Authors are thankful the reviewer for the suggestion, and the source. We have added the subsection 8.5 Molecular chaperones and cancer immunotherapy in the main manuscript as per the suggestion, also used the given DOIs for making this subsection. Track changed in the main manuscript (Line No. 484-532).
8.5 Molecular chaperones and cancer immunotherapy
The molecular identification of a large number of antigens that are linked with tu-mours has offered targets for the creation of new immunotherapies for the treatment of cancer [133]. The implications that molecular chaperones play in tumour immunity, as well as the useful features of molecular chaperones in cancer therapy, have garnered a growing proportion of interest in recent years [134, 135]. One can classify a chaperone ac-cording to one of these three tenets: The first advantage is that chaperones have the capac-ity to bind to a wide variety of peptides and proteins that are connected with tumours [136]. The second factor is the presence of particular receptors on the surfaces of anti-gen-presenting cells (APCs), which makes it possible for effective uptake of chaperones that are combined with peptides and proteins [137, 138]. Thirdly, chaperones interact with and activate innate immune components (such as APCs or NK cells), which assists in the initiation of adaptive immunological responses (such as the activation of CD8+ CTL and CD4+ T helper cells) [139, 140]. Heat shock proteins, often known as HSPs, are a type of molecular chaperone that are evolutionarily conserved but have a variety of roles in dif-ferent physiological processes [141]. The molecular sizes of the HSPs are used to catego-rise them into the following groups: HSP27, HSP40, HSP60, HSP70, and HSP90 [142]. By preserving the native folding energetics of the proteins, the HSPs inhibit the nonspecific aggregation of proteins within the cell. HSPs are effective biomarkers for assessing the stage of some types of cancer as well as the severity of the disease [143]. HSPs have been shown to have a role in the proliferation of tumour cells, as well as their differentiation, invasion, and metastasis [144]. The levels of HSP27 and HSP70 expression in tumour cells were found to have an effect on how well the cells responded to traditional cancer treat-ment [145]. Another important heat shock chaperonin protein, HSP60, is mostly found in mitochondria, where it contributes to the folding and transport of mitochondrial proteins [146]. HSP60 is a heat shock chaperonin protein. The high-molecular weight HSP90 chaperone is a crucial regulator of the process of tumour growth, similar to the other HSPs [147]. Not only are HSPs involved in the growth of tumours, but they also play a role in determining how they react to treatment. There is growing evidence that HSPs can be ef-fective targets for cancer immunotherapy [144, 148]. SRECI binds to a much wider variety of common heat shock proteins than LOX-1 does, including HSP60, HSP70, HSP90, HSP110, gp96, and GRP170 [149]. The majority of LOX-1’s binding occurs with HSP60 and HSP70. It is essential for immunosurveillance for peptides to be cross-presented in this manner because, not only is the attached peptide protected from degradation, but the efficiency of cross-presentation is also increased in dendritic cells [150]. Some heat shock proteins, such as HSP70 and HSP90, are also implicated in the intracellular cytosolic pathway of cross-presentation and the transportation of antigens from the endosome into the cytosol [140]. This process allows antigens to be transported from the endosome into the cytosol. Glucose regulated protein 94 (GRP94), also known as GP96, is a heat shock protein (HSP) family member that acts as a stress-inducible molecular chaperone [151]. GRP94 plays an essential part in the regulation of the delicate balance that exists between the survival and death of cancer cells [152]. Additionally, GRP94 is important for the chaperoning of many proteins, some of which have been shown to have essential roles in immunological response and in the genesis of cancer [135]. GRP94 protein has significant potential as both a biomarker and a therapeutic target. It has come to light that GRP94 is involved in the process of survival signalling by way of its client protein network, as well as the induction of the UPR and the modulation of the immune response. There is still po-tential for a targeted treatment that involves the selective suppression of GRP94. To this day, the GRP94 client network in its entirety has not been uncovered [153].
- Typographical errors can be avoided. The language and grammar used throughout the manuscript need to be improved. Specific attention needs to be given to this which will improve the standard of the manuscript.
Response 5: We have corrected the grammatically error throughout the manuscript, track changed in the main manuscript.
- Please improve the conclusion with clear Future perspectives and strategies with more emphasis on immunotherapy and other upcoming cancer therapy.
Response 6: We have improved the conclusion part focusing on strategies with more emphasis on immunotherapy and other upcoming cancer therapy. We have satisfied the comment, track changed in the main manuscript (Line No. 570-586).
Immunotherapy is an exciting new type of advanced cancer treatment that has the potential to be a major improvement in the management of currently available tumours. The future should be focused on the restoration of specific immune immunosuppressive pathway in the anti-tumor process, rather than simply enhancing the broad and untar-geted systemic immune response. This should include the following three principles: de-ciding that the immunosuppressive microenvironment is caused by the tumour, focusing the immunosuppression on the tumour microenvironment, and finding new targets act-ing on the main functional pathways [10, 163]. Therefore, the goal of cancer immuno-therapies is to clarify the molecular and cellular mechanism by which cancer cells can dodge the immune system, resulting to therapeutic interventions that increase anti-tumor immunity [164]. Cancer immunotherapies have been shown to be effective in treating pa-tients with advanced stages of the disease. It is anticipated that upcoming developments in cancer immunotherapy will overcome and address a significant number of these prob-lems. The development of more therapeutic targets, personalised biomarker profiles, drug combination therapies that will strengthen efficacy and reduce toxicity, and immunopre-ventive techniques that will lessen cancer rates and relapse as well as the affiliated treat-ment costs are expected to be among the anticipated innovations [7].